# The Role of Imaging in Inflammatory Bowel Diseases: From Diagnosis to Individualized Therapy

**DOI:** 10.3390/diagnostics15192457

**Published:** 2025-09-26

**Authors:** Salvatore Lavalle, Alessandro Vitello, Edoardo Masiello, Giuseppe Dell’Anna, Placido Romeo, Angelo Montana, Giambattista Privitera, Michele Cosenza, Domenico Santangelo, Tommaso Russo, Federico Bonomo, Emanuele Sinagra, Partha Pal, Antonio Facciorusso, Fabio Salvatore Macaluso, Ambrogio Orlando, Marcello Maida

**Affiliations:** 1Faculty of Medicine and Surgery, University of Enna “Kore”, 94100 Enna, Italy; salvatore.lavalle@unikore.it (S.L.);; 2Radiology Unit, Umberto I Hospital, 94100 Enna, Italy; 3Gastroenterology Unit, Umberto I Hospital, 94100 Enna, Italy; 4Department of Radiology, IRCCS San Raffaele Hospital, 20132 Milan, Italy; masiello.edoardo@hsr.it (E.M.); santangelo.domenico@hsr.it (D.S.); russo.tommaso1@hsr.it (T.R.); 5Gastroenterology and Gastrointestinal Endoscopy Unit, IRCCS Policlinico San Donato, 20097 Milan, Italy; dellanna.giuseppe@hsr.it; 6Division of Radiology, San Marco Hospital, AOU Policlinico “G. Rodolico” San Marco, 95121 Catania, Italy; 7Department of Advanced Biomedical Sciences, University of Naples Federico II, 80131 Naples, Italy; 8Gastroenterology Unit, Fondazione Istituto San Raffaele Giglio, 90015 Cefalù, Italy; 9Medical Gastroenterology, Asian Institute of Gastroenterology, Somajiguda, Hyderabad 500082, India; 10Gastroenterology Unit, Department of Experimental Medicine, University of Salento, 73100 Lecce, Italy; 11Clinical Effectiveness Research Group, University of Oslo, 0313 Oslo, Norway; 12IBD Unit, ‘Villa Sofia-Cervello’ Hospital, 90146 Palermo, Italy; fsmacaluso@gmail.com (F.S.M.); ambrogiorlando@gmail.com (A.O.)

**Keywords:** inflammatory bowel diseases, Crohn’s disease, ulcerative colitis, microbiota, intestinal ultrasound, computed tomography enterography, magnetic resonance enterography, artificial intelligence, biological therapy

## Abstract

Background: Inflammatory Bowel Disease (IBD), comprising Crohn’s disease and ulcerative colitis, requires accurate assessment over time. Imaging techniques play a crucial role in diagnosis, monitoring disease activity, and guiding therapeutic response. This review summarizes the current evidence on radiologic imaging techniques in IBD, focusing on intestinal ultrasound (IUS), computed tomography enterography (CTE), magnetic resonance enterography (MRE), and other emerging technologies. Methods: A literature review was conducted using PubMed, EMBASE, Scopus, and the Cochrane Library, encompassing publications up to 31 October 2024. Results: IUS offers a non-invasive tool for assessing bowel wall thickness, vascularity, and complications. CTE and MRE provide detailed visualization of luminal and extraluminal disease, with MRE preferred for routine monitoring due to the absence of ionizing radiation. Standardized indices and scoring systems aid in objective disease activity assessment. Emerging technologies like Positron Emission Tomography (PET)/MRI and radiomics show promise in combining metabolic and morphological information for complex cases. Conclusions: Imaging has a central role in IBD management, with IUS, CTE, and MRE demonstrating high diagnostic accuracy. Radiomics and Artificial Intelligence (AI) are paving the way for precision imaging. Integrating advanced imaging techniques, scoring systems, and AI-driven analytics represents a transformative step toward more effective and individualized care for patients with IBD.

## 1. Introduction

Inflammatory Bowel Disease (IBD) encompasses a group of chronic inflammatory disorders primarily affecting the gastrointestinal tract, characterized by an abnormal immune response. The two main types of IBD, Crohn’s disease (CD) and ulcerative colitis (UC), have transitioned from being predominantly Western ailments to global health concerns [1]. The age distribution of IBD is typically bimodal, with a primary peak between 20–40 years and a secondary peak after 60 years. Notably, pediatric-onset IBD now constitutes up to 25% of new diagnoses [2]. Despite extensive research, the precise etiology of IBD remains unclear [1]; however, current evidence strongly indicates a multifactorial origin involving genetic predisposition, environmental triggers, gut microbiota alterations, and dysregulated immune responses [3]. Given the chronic and relapsing nature of IBD, management is centered around achieving and maintaining remission, preventing complications, and improving patient’s quality of life. Therapeutic strategies include anti-inflammatory drugs (aminosalicylates, corticosteroids), immunosuppressive agents (azathioprine, 6-mercaptopurine, methotrexate), biologics targeting specific inflammatory pathways (anti-TNF, anti-integrins, anti-IL-12/23 agents or selective IL-23 inhibitors), new advanced oral treatments like JAK inhibitors (Tofacitinib, Filgotinib, Upadacitinib) and S1P1R modulators (Ozanimod and Etrasimod), dietary interventions, and, when necessary, surgical procedures [4,5]. The diagnosis of IBD requires a multifaceted approach that integrates clinical, biochemical, endoscopic, histological, and radiological findings [6]. While ileocolonoscopy with biopsy remains the gold standard for initial diagnostic confirmation and disease classification, cross-sectional imaging techniques are essential in characterizing disease extension, identifying transmural or extramural complications, and assessing segments inaccessible to endoscopy [7]. Intestinal ultrasound (IUS), Magnetic Resonance Enterography (MRE) and Computed Tomography (CT) providing accurate evaluation of small bowel involvement are indispensable in detecting strictures, fistulas, and abscesses [8]. These tools do not just help at the beginning: they are also essential for monitoring how well treatment is working overtime [9,10]. This review is primarily intended for both gastroenterologists and radiologists, reflecting the increasing need for a shared language and integrated approach in the management of IBD. For clinicians, it aims to clarify the principles and clinical indications of the main imaging modalities, while for radiologists and sonographers it provides technical details related to image acquisition and interpretation, bridging the gap between the technical aspects of imaging and its clinical applications.

## 2. Materials and Methods

This review aimed to consolidate current evidence on the role of radiologic imaging techniques in the diagnosis, assessment of disease activity, and monitoring of IBD, encompassing both CD and UC. A comprehensive literature search was conducted across PubMed, EMBASE, Scopus, and the Cochrane Library databases, covering publications up to 31 July 2025. The search strategy utilized Medical Subject Headings (MeSH) and relevant keywords, including “Intestinal Ultrasound,” “Bowel Ultrasonography,” “Magnetic Resonance Enterography,” “Computed Tomography Enterography,” “Inflammatory Bowel Disease,” “Crohn’s Disease,” “Ulcerative Colitis,” “Disease Activity,” “Treatment Response,” “Remission,” and “Transmural Healing.” Boolean operators “AND” and “OR” were employed to refine the search results. Inclusion criteria encompassed observational studies, clinical trials, and systematic reviews that evaluated the application of IUS, MRE, or CT in assessing disease activity, treatment response, or remission in IBD patients. Studies focusing on pediatric populations, or postoperative assessments were also considered. Exclusion criteria included non-English publications, case reports, editorials, and studies lacking sufficient methodological detail. Data extraction was performed independently by two reviewers, focusing on study characteristics (authors, year, design), patient demographics, imaging parameters assessed, reference standards used, and key findings related to the diagnostic accuracy, monitoring capabilities, and prognostic value. Discrepancies were resolved through discussion or consultation with a third reviewer. The quality of included studies was appraised using appropriate tools: the Quality Assessment of Diagnostic Accuracy Studies-2 (QUADAS-2) for diagnostic studies and the Cochrane Risk of Bias tool for randomized controlled trials.

## 3. Results

### 3.1. Clinical and Endoscopic Scoring Systems in IBD

Accurate assessment of disease activity in IBD is crucial for effective patient management and therapeutic decision-making. To provide a more practical overview, we have included an algorithm summarizing the optimal timing and indications of different imaging modalities in IBD, integrating current ECCO–ESGAR and AGA recommendations (Figure 1).

Clinical scoring systems [11,12,13,14,15,16,17], such as the Crohn’s Disease Activity Index (CDAI) [15] and Harvey–Bradshaw Index (HBI) [16] for CD, as well as the partial Mayo Score [12] for UC, integrate patient-reported symptoms, physical examination findings, and laboratory results to provide an overview of disease activity. These tools facilitate standardized assessments, enabling comparisons across clinical trials and clinical practice, helping stratify patients for suitable therapies. Endoscopic scoring systems [12,18,19,20,21], including the Simple Endoscopic Score for Crohn’s Disease (SES-CD) [20] for CD, the Mayo Endoscopic Sub-score (MES) [12] and the Ulcerative Colitis Endoscopic Index of Severity (UCEIS) [18] for UC, offer direct visualization and evaluation of mucosal inflammation and healing. These indices are instrumental in assessing the efficacy of therapeutic interventions and have become integral endpoints in clinical research (Table 1).

### 3.2. Ultrasound

IUS has emerged as a powerful, non-invasive tool for the assessment and monitoring of patients with IBD, particularly CD, but increasingly also UC [24,25]. IUS enables real-time visualization of bowel wall thickness, vascularity, and extramural complications such as fistulas and abscesses [26]. In this context, Color Doppler ultrasound provides a semi-quantitative evaluation of mural and perimural vascularity, with increased signal reflecting hyperemia and correlating with active inflammation. In addition, contrast-enhanced ultrasound (CEUS), by means of intravenous microbubble contrast agents, allows a more sensitive and quantitative assessment of bowel wall perfusion. According to the latest ECCO-ESGAR guidelines, IUS demonstrates a diagnostic accuracy comparable to MRE, with sensitivity and specificity values reaching 94% and 97%, respectively for small bowel CD [27]. Calabrese et al. reported that IUS correlate well with endoscopic and clinical indices of disease activity, and its reproducibility makes it suitable for repeated use in both outpatient and emergency settings [28]. In this study, IUS demonstrated a sensitivity of 79.7% and a specificity of 96.7% for diagnosing suspected CD, and a sensitivity of 89% with a specificity of 94.3% in patients with known CD, particularly when oral contrast agents are used to enhance visualization [28]. While IUS is most effective in detecting inflammation in the terminal ileum and sigmoid colon, its performance decreases in the rectum due to anatomical limitations; however, the addition of transperineal ultrasound can improve the assessment of perianal disease [29]. A UK multicenter trial involving 284 CD patients demonstrated that although MRE had slightly higher sensitivity for disease extent (80% vs. 70%), IUS had high sensitivity (92%) and specificity (81%) for detecting active small bowel involvement, affirming its utility in both newly diagnosed and relapsed cases [30]. IUS is increasingly recognized not only for disease assessment but also for monitoring response to therapy. In fact, it has emerged as an effective tool for predicting outcomes of biologic treatments, particularly in CD. Several studies underscore its ability to differentiate responders from non-responders early, anticipate endoscopic improvement or remission, and assess long-term disease progression [31,32,33,34,35,36,37,38,39,40,41,42,43]. Key IUS parameters, such as bowel wall thickness (BWT) and color Doppler signal (CDS), have shown strong prognostic value in this setting [31,32,33,34,35,36,37,38,39,40,41,42,43]. With regard to UC, a prospective study involving 83 patients with moderate-to-severe UC showed a strong concordance (κ = 0.76–0.90) between IUS and endoscopic scores during follow-up at 3, 9, and 15 months, supporting its use as a surrogate marker for mucosal healing (MH) [44]. Furthermore, IUS has demonstrated utility in special populations, including pregnant patients and those with significant comorbidities, where exposure to radiation or contrast agents is undesirable. Despite limitations, such as operator dependence and reduced sensitivity in proximal small bowel or rectal disease, IUS is increasingly recognized as a reliable and patient-centered modality, now integrated into treat-to-target strategies and recommended by recent AGA guidance [45]. Beyond ECCO-endorsed data, Ripollés et al. emphasized the utility of contrast-enhanced ultrasound (CEUS) in characterizing stenosis (fibrotic vs. inflammatory) and assessing bowel wall vascularization, thereby providing an early indicator of therapeutic response [46]. Moreover, Novak et al. demonstrated that bowel wall thickness and Doppler signal detected via IUS correlate strongly with clinical relapse in CD, thus offering predictive value [47]. A systematic review and meta-analysis by Bots et al. concluded that IUS is highly accurate for assessing disease activity, with pooled sensitivity and specificity of 85% and 91%, respectively, across multiple cohorts [48]. Another study by Kucharzik et al. confirmed the feasibility of point-of-care IUS as an extension of physical examination in daily clinical practice, promoting its role in tight monitoring protocols [49]. The utility of IUS in the setting of postoperative recurrence (POR) in CD patients has been extensively investigated, highlighting its value as a non-invasive alternative to endoscopy [50]. Rispo et al. performed a meta-analysis indicating that a BWT ≥ 5.5 mm may be effective for identifying severe POR (defined as Rutgeerts score ≥ 3), with a pooled sensitivity of 83.8% and specificity of 97.7% [51]. A recent multicenter prospective study evaluated 91 operated CD patients within 1-year post-surgery using both IUS and colonoscopy. On note, the combination of a BWT ≥ 3 mm and fecal calprotectin (FCP) levels > 50 μg/g identified 75% of individuals with endoscopic recurrence, while maintaining a low false positive rate of less than 5% [52]. Given its cost-effectiveness, safety, and accessibility, IUS is increasingly integrated into treat-to-target strategies for IBD management. Its expanding role is particularly valuable in pediatric populations, where repeated endoscopy or radiological imaging is less feasible. While the present review does not primarily aim to analyze pediatric populations recent studies have underscored the diagnostic value of IUS in pediatric IBD, showing strong correlation with endoscopic and histologic findings. Conventional B-mode ultrasound demonstrates a sensitivity of 74–88% and specificity of 78–93% for detecting terminal ileal Crohn’s disease [43].

#### Intestinal Ultrasounds Imaging Findings and Scoring Systems

One of the primary ultrasonographic findings in IBD is BWT [53]. In CD, a BWT greater than 3 mm is often indicative of active inflammation, correlating well with endoscopic and histological findings [54]. Additionally, the preservation or loss of bowel wall stratification provides insights into disease activity; loss of stratification is associated with severe inflammation and may predict a higher risk of surgical intervention [55]. The assessment of bowel wall stratification (BWS) is another important parameter. Normally, the bowel wall exhibits a five-layered structure with alternating echogenicity. Loss of this stratification suggests active transmural inflammation, a feature more prevalent in CD due to its transmural nature [56]. In UC, stratification is typically preserved unless in severe cases [57]. Color Doppler imaging enhances the assessment by evaluating vascularity within the bowel wall [58]. The ultrasound score developed in 1994 by German physician B. Limberg is widely used to semi-quantitatively assess this parameter (along with BWT), by categorizing visible vascularization into four grades, with higher scores reflecting increased disease activity [59]. Mesenteric fat proliferation and lymphadenopathy are additional findings frequently observed in CD, reflecting chronic inflammation and aiding in disease assessment [60]. IUS is also adept at identifying complications such as strictures, fistulas, and abscesses, particularly in CD. Its ability to detect these features in real-time facilitates prompt clinical decision-making [26]. In UC, while the disease primarily affects the mucosal layer, IUS can still provide valuable information regarding disease extent and activity, especially in severe cases where transmural involvement may occur [61,62]. The ECCO and the ESGAR endorse the use of IUS for both initial assessment and ongoing monitoring of IBD [27,63]. Among the various scoring systems developed to quantify IUS findings, several have gained prominence due to their validation against endoscopic standards and clinical applicability. The International Bowel Ultrasound Segmental Activity Score (IBUS-SAS) stands out due to its comprehensive approach and robust validation. IBUS-SAS evaluates four key parameters: BWT, BWS, CDS, and inflammatory fat (i-fat) [47]. In a retrospective study involving 203 CD patients, IBUS-SAS demonstrated a strong correlation with the SES-CD, yielding a correlation coefficient (r) of 0.891 (*p* < 0.001). Additionally, it showed significant correlations with clinical indices like the CDAI (r = 0.590, *p* < 0.001) and biomarkers such as C-reactive protein (CRP) (r = 0.688, *p* < 0.001) [64]. Further validation came from a study assessing the diagnostic performance of IBUS-SAS in 140 CD patients. The study reported an area under the curve (AUROC) of 0.895, with a sensitivity of 85.4% and specificity of 82.4% at a cutoff value of 48.7. IBUS-SAS also showed significant correlations with SES-CD (r = 0.511), CDAI (r = 0.666), and CRP (r = 0.645), outperforming the Simple Ultrasound Score for Crohn’s Disease (SUS-CD) in these metrics [65]. The Simple Ultrasound Score for Crohn’s Disease (SUS-CD) offers a more streamlined assessment, focusing on BWT and CDS [64]. A multicenter Spanish study validated SUS-CD against SES-CD, reporting an AUROC of 0.923, with 90% sensitivity and 86.4% specificity at a cutoff score of 5.5 [66]. Another study highlighted that SUS-CD scores greater than 1 accurately detected any endoscopic activity (SES-CD > 2), while scores above 3 were indicative of moderate activity (SES-CD > 7), with AUROCs of 0.92 and 0.88, respectively [67]. For UC, the Ulcerative Colitis Intestinal Ultrasound Index (UC-IUS) has been developed to correlate IUS findings with endoscopic disease activity. Furthermore, in pediatric populations, UC-IUS demonstrated high sensitivity and specificity in detecting moderate-to-severe disease activity, outperforming other indices like the Civitelli index [68]. Recently, the Milan Ultrasound Criteria (MUC), a newly developed and validated ultrasound score for adult UC patients [25], has shown greater accuracy than the MES in predicting the risk of colectomy [69]. This was demonstrated in a large study of 141 patients with moderate-to-severe UC, 13 of whom required colectomy due to refractoriness to medical therapy [69]. In conclusion, IUS scoring systems, as summarized in Table 2, offer standardized, non-invasive metrics for evaluating disease activity in CD and UC, enabling reliable monitoring and alignment with clinical and endoscopic findings.

### 3.3. CT and CT Enterography

CT and CT Enterography (CTE) are indispensable imaging modalities in the assessment of IBD, particularly CD, due to their high spatial resolution, speed, and capacity to detect transmural inflammation, strictures, and extraluminal complications such as fistulas and abscesses [70]. While ileocolonoscopy remains the gold standard for initial diagnosis and surveillance of mucosal disease, cross-sectional imaging plays a critical complementary role, especially in the small bowel, which is less accessible endoscopically [71]. According to the ECCO-ESGAR consensus guidelines, CT and MRI are both highly sensitive and specific for the evaluation of small bowel lesions and disease extent, although MRI is generally preferred for routine follow-up due to the absence of ionizing radiation, which is particularly important in the typically young IBD population [27]. The acquisition protocol combines ingestion of a large volume of oral contrast to distend the bowel with intravenous administration of contrast material to enhance visualization of the bowel wall, mucosal patterns, and extramural complications [72] (Table 3). CTE enhances traditional CT by combining enteric contrast agents and thin section multidetector CT imaging, allowing for superior bowel wall visualization and detection of mural hyperenhancement, fat stranding, comb sign, and other radiological features of active inflammation [73]. CTE has demonstrated high sensitivity and specificity in detecting active small bowel inflammation. A study reported a sensitivity of 93.88% and specificity of 85.71% for CTE in assessing small bowel CD, with an overall diagnostic accuracy of 92.06% [74]. In a comparative study, the sensitivity and specificity for diagnosing terminal ileum CD were 76% and 85% for CTE, respectively. While capsule endoscopy showed higher sensitivity (100%) and specificity (91%), CTE remains a valuable tool, especially when capsule endoscopy is contraindicated [75]. CT is also especially useful in detecting complications such as penetrating disease, abscesses, and fistulas that may not be clinically apparent [76]. In a study involving 36 patients, CTE accurately determined the presence or absence of strictures, fistulas, abscesses, or inflammatory masses with accuracies of 100%, 94%, 100%, and 97%, respectively [77]. Despite its strengths, CT-based imaging is limited by cumulative radiation exposure, making its repeated use less desirable in chronic conditions like IBD. As such, it is typically reserved for acute settings (e.g., suspected perforation or abscess), initial staging in complex disease, in elderly patients (due to its speed, and the perceived lower risk of radiation exposure compared to younger individuals) or cases where MRI is unavailable or contraindicated (e.g., patients with incompatible metallic implants or severe claustrophobia). Nevertheless, ECCO-ESGAR recommends that all newly diagnosed CD patients undergo small bowel assessment through imaging, and CTE is a preferred modality where MRI access is limited [27]. In addition to diagnosis, CTE is valuable in monitoring treatment response. Several studies have shown that imaging features such as reduction in wall thickness and contrast enhancement correlate well with clinical remission and endoscopic MH. For instance, bowel wall hyperenhancement and stratification, when absent, may indicate deep remission, an important emerging therapeutic goal in IBD management. A study by Wu et al. evaluated 50 CD patients and found that post-treatment CTE showed significant reductions in bowel wall thickness and attenuation. Specifically, bowel wall thickness decreased from 8.8 ± 2.8 mm to 6.4 ± 1.9 mm, and attenuation values dropped from 90.0 ± 15.4 HU to 73.4 ± 14.2 HU. These imaging changes were strongly associated with clinical remission, indicating that CTE can effectively monitor therapeutic response [78]. Tong et al. introduced the CT Enterography Index of Activity (CTEIA), a quantitative scoring system derived from CTE findings such as mural thickness, stratification, and the comb sign. This index showed a high correlation (r = 0.779, *p* < 0.001) with the CDEIS, suggesting that CTEIA can reliably assess disease activity and monitor treatment efficacy [79]. A prospective study conducted by Lopes et al. assessed the relationship between CTE findings and endoscopic activity in newly diagnosed CD patients. After 1 year of immunosuppressive therapy, improvements in CTE features like mural hyperenhancement, mesenteric fat densification, and the comb sign were significantly associated with endoscopic remission [80]. Hara et al. conducted a preliminary study demonstrating that changes observed in sequential CTE examinations correlated well with disease activity in CD patients. This finding supports the utility of CTE in tracking disease progression or remission over time [81]. In conclusion, CT and CTE remain vital tools in the comprehensive management of IBD. Their ability to provide rapid, detailed, and reproducible visualization of both luminal and extraluminal disease complements clinical, endoscopic, and biochemical assessments. Their judicious use, particularly when MRI is unavailable, offers a powerful approach for assessing disease extent, guiding therapeutic strategy, and identifying complications that directly impact prognosis and quality of life. Future directions involve optimizing low-dose CT protocols and integrating radiomic analysis to enhance non-invasive disease characterization [82].

### 3.4. Magnetic Resonance Imaging

MRI has emerged as a crucial non-invasive tool for evaluating the bowel wall and extraluminal features of IBD. MRE allows detailed assessment of small-bowel CD and related complications, complementing endoscopy by visualizing transmural and extramural disease components [83]. Current consensus guidelines recommend performing cross-sectional imaging (CT or MRI) at the time of CD diagnosis to map disease extent, and to consider MRI for routine monitoring of small-bowel CD activity over time [83]. The main MRI sequences used in the evaluation of IBD, together with their technical principles, imaging characteristics, and clinical relevance, are summarized in Table 4.

#### 3.4.1. MRI Techniques and Sequences in IBD

Dedicated MRI protocols have been developed for evaluating IBD, often termed MR enterography (when oral contrast is used) or MR enteroclysis (when contrast is delivered via nasojejunal tube). Key aspects of MRI technique include adequate bowel distension, optimal sequence selection, and motion artifact reduction. Patients are typically fasted 4–6 h prior, then given oral hyperosmolar contrast (e.g., mannitol, polyethylene glycol) to distend the small bowel [84]. Studies show that lack of oral contrast significantly reduces diagnostic accuracy, so ingestion of ~1000 mL of contrast solution about 45 min before scanning is recommended [85,86]. Intravenous antispasmodic agents (e.g., hyoscine butylbromide or glucagon) are routinely administered to reduce peristalsis, which improves small-bowel distension and image quality [87].

Typical MRI sequences employed for the evaluation of IBD are summarized in Table 5. For MR colonography (imaging focused on the colon), an additional rectal enema with water or contrast can be administered to better distend the colon. However, ECCO-ESGAR guidelines note that routine MRE for CD typically does not require dedicated colonic preparation, except in specific cases [27,84]. When imaging perianal CD, a dedicated MRI of the pelvis is performed using high-resolution T2-weighted images (in orthogonal planes aligned to the anal canal) with and without fat saturation [88]. Intravenous contrast on T1 images is helpful in perianal fistula MRI to distinguish active fistulous tracts or abscess (enhancing granulation tissue) from simple fluid collections [89]. Overall, adherence to standardized MRI techniques and adequate training improves diagnostic yield. Radiologists generally achieve significantly better accuracy after interpreting > 100 MRE cases, underscoring the importance of experience and training in MRI reading for IBD [90].

#### 3.4.2. Typical MRI Findings in Crohn’s Disease

On MRI, active CD classically shows bowel wall thickening (often >5 mm) with mural edema and contrast enhancement. The bowel wall may exhibit a layered enhancement pattern on post-contrast images, termed stratified mural hyperenhancement, reflecting inflamed mucosa and submucosa with relative sparing of deeper layers (submucosal edema or fat can cause a bilaminar or trilaminar appearance) [83]. Active CD inflammation often causes loss of the normal layered wall architecture on MRI, especially in severe cases, due to transmural ulceration and edema [91]. T2-weighted images show high signal in the thickened bowel wall from edema (and can even depict ulcers as luminal T2 bright spots or fissures). Surrounding the affected bowel, mesenteric changes are common: the “comb sign” refers to engorged vasa recta running to the diseased bowel segment, which is well-seen on post-contrast MRI [92]. The mesenteric fat adjacent to active segments often appears inflamed (edematous or fibrofatty proliferation, known as “creeping fat”). Enlarged mesenteric lymph nodes may also be present [93]. A hallmark of CD is its propensity for transmural complications. MRI is very sensitive for detecting fistulas (abnormal sinus tracts connecting thbowel to other bowel loops or organs). On MRI they appear as T2-hyperintense tracks extending from the bowel wall, often with enhancement if active. For example, entero-enteric or entero-colic fistulas may be seen as enhancing connections between loops, sometimes with multiple interconnected loops creating a “star” or asterisk-shaped pattern in the mesentery [94]. Another common complication is stricture formation (fibrostenotic disease). An MRI-defined stricture is present when there is a narrowed intestinal lumen at a diseased segment with proximal bowel dilation > 3 cm or significant stasis [95]. Active inflammatory strictures typically show thick enhancement and edema in the wall, whereas chronic fibrotic strictures may show a narrower caliber with less edema. MRI can identify strictures and signs of sub-obstruction (pre-stenotic dilation, “small-bowel feces” sign upstream) [96]. It is worth noting that, to differentiate fibrotic from active inflammatory strictures, some authors recommend the use of the ‘cine loop’ technique. In this approach, inflammatory strictures typically show dynamic opening on cine imaging, whereas fibrotic strictures remain fixed on multiple pulse sequences, fluoroscopic observation, or serial imaging examinations [83,97]. Intra-abdominal abscesses can arise from penetrating CD; on MRI, abscesses manifest as fluid or pus collections with T2 hyperintensity and rim enhancement, often adjacent to a fistula or perforated loop [98].

#### 3.4.3. Typical MRI Findings in Ulcerative and Indeterminate Colitis

UC is a colonic-limited disease characterized by continuous mucosal inflammation beginning in the rectum and extending proximally. Because UC inflammation is typically superficial (mucosa/submucosa) and continuous, MRI findings differ from CD. Bowel wall thickening in UC is generally less pronounced than in CD’s colitis. Studies report an average colonic wall thickness around 7–8 mm in active UC, which is less than the often-greater thickness seen in CD’s colitis [99].

MRI may show uniform, symmetric wall thickening in the affected colonic segment, with enhancement primarily of the mucosa in moderate disease. Stratified enhancement can also be seen in UC during active flares (due to submucosal edema and hyperemia), but the fat wrapping and comb sign that are common in CD are typically absent in UC [100]. Another distinguishing feature is the lack of skip lesions: MRI of UC will show a contiguous segment of colitis (e.g., pancolitis involving the entire colon), whereas intervening normal bowel strongly suggests CD. UC also rarely forms fistulas or abscesses; thus, finding a fistula tract on MRI virtually excludes pure UC and points to CD [101]. In chronic longstanding UC, a “fat halo” (submucosal fat deposition) in the colonic wall may be seen on CT/MRI, reflecting chronic inflammation and steroid use—this can help indicate chronic UC changes [102]. MRI in acute severe UC can demonstrate marked colonic wall edema, enhancement, and even toxic megacolon (gross distension with thinning of the wall and possible perforation). Diffusion-weighted MRI has shown utility in UC as well: active inflammation in UC results in restricted diffusion in the colonic wall, which can highlight inflamed segments [103].

Overall, MRI is highly effective for CD evaluation due to its ability to image the entire bowel wall and beyond. In UC, MRI is somewhat less critical for routine assessment (since disease is superficial and confined to colon, often well-monitored by colonoscopy), but it can be valuable in severe flares or when an alternate diagnosis is suspected. MRE has shown good accuracy even in UC: one prospective study reported about 87% sensitivity and 88% specificity of MRI (with colon distension) for detecting active UC, using endoscopy as the reference [104]. This demonstrates that MRI can depict colonic inflammation reasonably well, especially when moderate-to-severe (wall thickness > 3–4 mm) [105]. Mild UC (limited to mucosal changes) may not cause enough mural change to be reliably seen on MRI, so endoscopy remains superior for mild disease detection [106].

#### 3.4.4. Diagnostic Performance of MRI vs. Endoscopy and Histology

Multiple studies and meta-analyses have evaluated the diagnostic performance of MRI for active IBD, comparing it to endoscopic and histologic standards. In CD, MRE has shown high accuracy for detecting active inflammation, particularly in the small bowel. A meta-analysis (in the era up to about 2018) found pooled sensitivity of approximately 95% and specificity around 92% for MRE in identifying active CD’s lesions in the small intestine [30].

Notably, MRI and IUS have comparable overall accuracy for active CDs in many studies, though MRI may be slightly more sensitive for deep small-bowel segments [107].

For colonic CD’s disease, MRI tends to have somewhat lower sensitivity. One study reported MRE sensitivity ~76% and specificity ~90% against colonoscopy for CD’s colitis, implying that MRI is very good at confirming colonic disease when positive (high specificity to “rule in” inflammation) but can miss some mild mucosal lesions (moderate sensitivity) [108].

Similarly in UC, as noted, MRI’s sensitivity dips for mild superficial disease. Endoscopy with biopsy can detect subtle mucosal erythema or microscopic inflammation that MRI cannot resolve. Histology remains the gold standard for confirming inflammation and assessing microscopic disease activity or remission. MRI cannot directly visualize microscopic changes, but it correlates strongly with macroscopic (endoscopic) disease activity. For example, segments with ulcers on endoscopy (indicating severe inflammation) almost always show corresponding MRI abnormalities (thickening, ulceration, high MaRIA score, etc.), and conversely, normal-appearing MRI segments generally lack active ulceration endoscopically [84].

#### 3.4.5. MRI-Based Scoring Systems in IBD

Standardized MRI-based indices have been developed to quantify IBD inflammation objectively, analogous to endoscopic indices like the CDEIS or SES-CD. The use of MRI indices allows radiologists and clinicians to grade disease activity and monitor changes over time in a reproducible way. Several validated MRI scores exist for luminal CD, of which the Magnetic Resonance Index of Activity (MaRIA) and its variants are the most established.

The MaRIA score is a segmental activity index originally derived by Rimola et al. to quantify CDseverity on MRE. MaRIA considers four MRI parameters on a per-bowel-segment basis: wall thickness, degree of contrast enhancement, presence of edema, and presence of ulceration [109]. Each parameter contributes to a sub-score for that segment. In validation studies, the MaRIA showed an excellent correlation with standard endoscopic indices, with a correlation coefficient r ≈ 0.83 between MaRIA and the CDEIS, indicating that higher MRI scores mirror more severe endoscopic disease [110]. A MaRIA segment sub-score ≥ 7 has been identified as the optimal cutoff for active CD (corresponding to mucosal lesions on endoscopy), and ≥11 indicates severe activity (usually correlating with the presence of deep ulcers on endoscopy) [8].

Using thresholds like these, MaRIA delivers high diagnostic performance: one study showed a segment MaRIA ≥ 11 detected ulcerations with ~78% sensitivity and 98% specificity and MaRIA ≥ 7 detected any active inflammation with ~87% sensitivity and 86% specificity [111]. MaRIA has become a reference standard in clinical trials for assessing MRI remission [112]. Its main limitation is that the original validation required rigorous bowel preparation including rectal enema for colonic distension, which is not always performed in routine practice. The Simplified Magnetic Resonance Index of Activity (sMaRIA) is a streamlined scoring system designed to assess CD activity using MRE. It evaluates four key features per bowel segment: wall thickness greater than 3 mm, mural edema, fat stranding, and the presence of ulcers. Each feature is scored dichotomously, simplifying the assessment process. In its initial validation, sMaRIA demonstrated high diagnostic accuracy. A score of ≥1 identified active disease with 90% sensitivity and 81% specificity (AUC = 0.91), while a score of ≥2 detected severe lesions with 85% sensitivity and 92% specificity (AUC = 0.94). The sMaRIA also showed strong correlations with established indices: r = 0.83 with the CDEIS and r = 0.93 with the original MaRIA score [113]. Further studies have corroborated these findings. For instance, a study reported that sMaRIA scores of ≥1 and ≥2 corresponded to sensitivities of 97.5% and 85%, and specificities of 97.3% and 92%, respectively, for detecting active and severe disease. The sMaRIA also correlated strongly with the SES-CD, with correlation coefficients ranging from r = 0.74 to r = 0.91 across different bowel segments [114]. Notably, the sMaRIA offers practical advantages over the original MaRIA score. It requires less time to calculate (approximately 4.5 min vs. 17 min) and does not necessitate gadolinium-based contrast agents, enhancing patient safety and comfort [115].

#### 3.4.6. MRI in Assessing Treatment Response

One of the valuable roles of MRI is evaluating how CD responds to therapy over time. The concept of “transmural healing (TH)” has gained attention: whereas endoscopic MH is defined by normalization of the intestinal lining, TH means resolution of inflammation through all layers of the bowel wall (and absence of fistulas/abscesses). MRI is uniquely suited to assess TH, since it can show improvements in wall thickness, edema, and enhancement that are invisible to endoscopy. Studies suggest that achieving TH using MRI may portend better long-term outcomes than MH alone [116].

In the context of monitoring IBD, MRI offers a noninvasive way to track disease without repeated endoscopies. For patients on therapy (like anti-TNF agents or newer biologics), periodic MREs can assess whether the bowel inflammation is regressing. For example, after 6–12 months of therapy, a baseline MaRIA of 15 might drop below 7 if deep remission is achieved, indicating radiologic healing [84].

MRI can also detect early signs of disease recurrence (e.g., new edema or ulcers in an asymptomatic patient post-surgery or in medically induced remission), sometimes before symptoms worsen. This could allow preemptive therapy adjustment. ECCO-ESGAR guidelines endorse the use of MRI for monitoring known CD, especially for the small bowel segments that are not accessible to colonoscopy [27,83,84].

However, it is also noted that a normal MRI does not always guarantee microscopic remission, so clinical judgment and complementary tests (like fecal calprotectin or limited endoscopy) are still used in combination.

### 3.5. PET/MR

Positron Emission Tomography combined with Magnetic Resonance Imaging (PET/MRI) has emerged as a promising hybrid imaging modality in the assessment of IBD, particularly CD, although it is not yet routinely used in clinical practice. By integrating the metabolic imaging capabilities of PET with the superior soft tissue contrast of MRI, PET/MRI provides comprehensive information on both the functional and structural aspects of bowel inflammation. In a study involving patients with CD, PET/MRI showed higher specificity in detecting active inflammation compared to PET alone, owing to MRI’s ability to provide detailed anatomical localization and reduce false positives associated with physiological uptake in the gastrointestinal tract [117].

Moreover, PET/MRI has been found to be more accurate than either modality alone in assessing disease activity. The simultaneous acquisition of PET and MRI data allows for precise spatial and temporal correlation, enhancing the detection of subtle lesions and extra-luminal disease manifestations. This is particularly beneficial in evaluating complex cases where conventional imaging modalities may fall short. In terms of diagnostic performance, PET/MRI has shown promising results. A study reported that PET/MRI had a sensitivity of 83% and an AUROC of 0.77 in detecting inflammation in the terminal ileum and colon, indicating its potential utility in clinical practice [118].

### 3.6. Radiomic and Artificial Intelligence in IBD Assessment

Radiomics and artificial intelligence (AI) are increasingly pivotal in the assessment of IBD, offering advanced tools for diagnosis, disease activity evaluation, and treatment monitoring. Radiomics involves extracting quantitative features from medical images, enabling the identification of patterns not discernible through conventional imaging analysis [9]. Recent studies have demonstrated the feasibility and effectiveness of radiomics-based models in differentiating between normal and abnormal intestinal ultrasound images in IBD patients. For instance, a radiomics-based classification model using XGBoost achieved an AUC of 0.98, with 93.8% sensitivity and specificity, outperforming convolutional neural network-based models. Similarly, in CTE, radiomics combined with machine learning algorithms has shown high accuracy in identifying CD lesions, with AUCs of 0.938 and 0.961 for arterial and venous-phase images, respectively [119].

Expanding beyond activity alone, newer work shows radiomics/AI may aid initial diagnosis and differential diagnosis. For instance, Zeng et al. developed a nomogram combining CTE radiomic signatures and clinical factors to stratify intestinal fibrosis in IBD, achieving validation AUC ~0.865, outperforming clinical-only models [120]. In MRI, radiomics has been utilized to distinguish between CD and UC, achieving an AUC of 0.874, indicating its potential in subtype classification. Moreover, AI models have been developed for histopathological analysis, classifying IBD activity grades in whole slide images with a weighted AUC of 0.871, enhancing consistency in disease activity assessment [121]. Lastly, a recent study developed a radiomics-based machine learning model to differentiate inflammation and fibrosis in stricturing CD using MRE. In a cohort of 51 CD patients, radiomics outperformed radiologist visual scoring, particularly in detecting severe fibrosis (AUC 0.78 vs. 0.35). Moreover, combining radiomics with radiologist assessment improved the overall diagnostic accuracy for both inflammation and fibrosis [122,123]. Laterza et al. demonstrated that CT-based radiomics could predict the 10-year risk of surgery in Crohn’s disease, offering prognostic stratification beyond conventional metrics [124]. Additionally, comprehensive reviews highlight AI models already being tested for predicting response to biologic therapy, risk of relapse, and distinguishing IBD from non-IBD colitides in early diagnostic stages [125,126]. While promising, these techniques still face limitations: many studies are retrospective and, single-center, with limited external validation and sometimes small sample sizes or lack of standardization. Nonetheless, the evidence now supports a potential role for AI/radiomics not only in monitoring disease, but also in early diagnostic differentiation and prognosis prediction in IBD. Moreover, several important limitations must be acknowledged to temper expectations and promote safe clinical translation. One major concern is the so-called “black box” problem, where many AI models—especially those based on deep learning—produce outputs without transparent or easily interpretable reasoning, which may hinder clinician trust and accountability [125,127]. Furthermore, performance of AI/radiomic models can degrade when applied to external or replication cohorts due to heterogeneity in imaging protocols, patient populations, scanner settings, and sample sizes; this raises risks of misdiagnosis or over-/under-prediction of disease in settings that differ from the development environment [127,128]. There is also the possibility of bias, for example, if training datasets are not representative (age, ethnicity, disease severity), which can lead to systematic errors. Finally, lack of regulatory oversight, insufficient external validation, and the need for models to be explainable and transparent through methods such as explainable AI (XAI) are essential to avoid harm when integrating AI into clinical decision-making [129].

A summary of the most relevant clinical scenarios in IBD and the corresponding preferred imaging modalities, with their respective strengths and limitations, is provided in Table 6. Figure 2, Figure 3, Figure 4 and Figure 5 illustrate multimodal imaging and endoscopic correlations across various clinical scenarios in patients with IBD, as evaluated at a tertiary care center.

## 4. Intestinal Capsule Endoscopy

Recent studies include the role of small bowel capsule endoscopy (SBCE) in reclassification of colonic IBD-U: Monteiro et al. applied SBCE in 36 IBD-U patients and found that 25% had significant small bowel inflammation (Lewis score ≥ 135), all of whom were later confirmed to have Crohn’s disease. The SBCE test had sensitivity 90%, specificity 100%, PPV 100%, and NPV 94%, demonstrating its value in differential diagnosis [130]. Hilmi et al. found with SBCE that a Lewis score cutoff ≥ 135 yielded diagnostic accuracy of ~83.2%, sensitivity ~89.5%, specificity ~78.9%, for detecting small bowel inflammatory activity, supporting the use of CE in suspected Crohn’s disease, especially when other modalities are inconclusive [131]. Additionally, in a pediatric cohort using pan-enteric capsule endoscopy, Oliva et al. demonstrated that the treat-to-target strategy based on capsule findings increased mucosal healing from 21% at baseline to 58% at week 52; also, management was altered in a large proportion of patients based on CE results [132].

## 5. Role of Imaging in Unclassified IBD

Unclassified IBD (IBD-U) refers to patients who present with features overlapping between CD and UC, such that after standard clinical, endoscopic, and histological evaluation no definitive classification is possible. Imaging modalities have increasingly been shown to aid in reclassification and prognosis in IBD-U. For example, the IBD-RADS system, derived from a large cohort (derivation: 606 patients, 365 CD vs. 241 UC; validation: 155 patients, 97 CD vs. 58 UC), showed that asymmetric enhancement, perianal fistulae, and visceral fat predominance (VFP) are independent predictors of CD. The system achieved an AUC of 0.929 in distinguishing CD from UC in the validation set (correctly classifying 98.0% of CD cases) using multiparametric imaging criteria on CT/MRI/CTE modalities [133]. Moreover, IUS has been evaluated in IBD-U and suspected IBD settings. The recent systematic review conducted by Hoffmann et al., found high sensitivity and specificity when IUS is used early in the disease course, including good discrimination in patients who had indeterminate features clinically; while precise numbers vary, B-mode bowel wall thickness, loss of stratification, and increased vascularity on Doppler correlate strongly with more advanced or transmural disease features, which may tip the balance towards a CD diagnosis over UC in IBD-U cases [134]. Finally, imaging also contributes prognostically in IBD-U. For patients initially unclassified, detection of skip lesions, transmural inflammation, or perianal disease via MRI or CT has been associated with later reclassification to Crohn’s disease and higher likelihood of requiring biologic therapy or surgical intervention. While numbers are still limited, these findings underscore a prognostic value that complements endoscopy and histology [135].

## 6. Malignancy Risk in IBD and the Role of Imaging

Patients with IBD, particularly those with extensive ulcerative colitis or colonic Crohn’s disease, carry a significantly elevated risk of colorectal neoplasia compared to the general population. Studies estimate that the standardized incidence ratio for colorectal cancer in ulcerative colitis lies between approximately 2–3× the baseline risk, especially after 8–10 years of disease duration, with risk increasing further in patients with severe or poorly controlled inflammation, backwash ileitis, strictures, or prior dysplasia [136,137].

Although colonoscopy with targeted biopsy and chromoendoscopy remains the gold standard for surveillance and detection of dysplasia, imaging modalities play an important complementary role. Cross-sectional imaging (MRI, CT) and, in certain settings, bowel ultrasound can help detect suspicious features that may indicate a neoplastic transformation, notably: asymmetric or mass-forming wall thickening, loss of normal mural stratification, irregular or heterogeneous contrast enhancement, luminal strictures with proximal dilatation, and enlarged mesenteric or regional lymph nodes [138,139]. MRI is well-suited to evaluate these features, offering good soft-tissue contrast, and the ability to assess both lumen and extraluminal involvement. MRI may identify lesions or suspicious changes beyond the reach of standard colonoscopy, particularly in the small bowel or proximal colon, or in patients with severe stricturing disease where direct visualization is compromised [138]. Additionally, recent guidelines emphasize that imaging should be considered in IBD patients who present with alarm features (e.g., anemia, change in bowel habit, unexplained weight loss), with stricturing disease, or in post-colectomy or pouch patients, to supplement endoscopic surveillance [140].

In summary, while endoscopic surveillance remains central for detection of dysplasia and CRC in IBD, imaging is an indispensable adjunct to identify neoplastic risk early, support differential diagnosis, and guide management decisions in high-risk subsets of patients.

## 7. Conclusions

In recent years, imaging has evolved from playing a supportive to a central role in the diagnosis, monitoring, and management of IBD. IUS, CTE and MRE, have demonstrated high diagnostic accuracy for detecting both luminal and extraluminal disease. Among these, IUS offers a non-invasive, patient-friendly, and cost-effective approach that enables the early assessment of treatment response. MRE, on the other hand, is distinguished by its superior soft tissue contrast and ability to evaluate transmural inflammation and complications, making it especially valuable for serial monitoring, particularly in younger patients. Standardized MRI-based indices such as MaRIA and sMaRIA, alongside novel scoring systems for ultrasound and CT, have provided objective, reproducible metrics for disease activity assessment, correlating closely with clinical and endoscopic indices. Moreover, emerging technologies such as PET/MRI have demonstrated the added value of combining metabolic and morphological information in complex IBD cases, while radiomics and AI are paving the way for precision imaging by extracting quantitative features to enhance diagnostic and prognostic capabilities. These advancements underscore a paradigm shift towards non-invasive, multi-parametric imaging approaches that support treat-to-target strategies and personalized therapy in IBD. Future research should aim to validate these tools in larger, multicenter cohorts and standardize imaging protocols for routine clinical use. Ultimately, the integration of advanced imaging techniques, scoring systems, and AI-driven analytics represents a transformative step toward more effective and individualized care for patients with IBD.

## Figures and Tables

**Figure 1 diagnostics-15-02457-f001:**
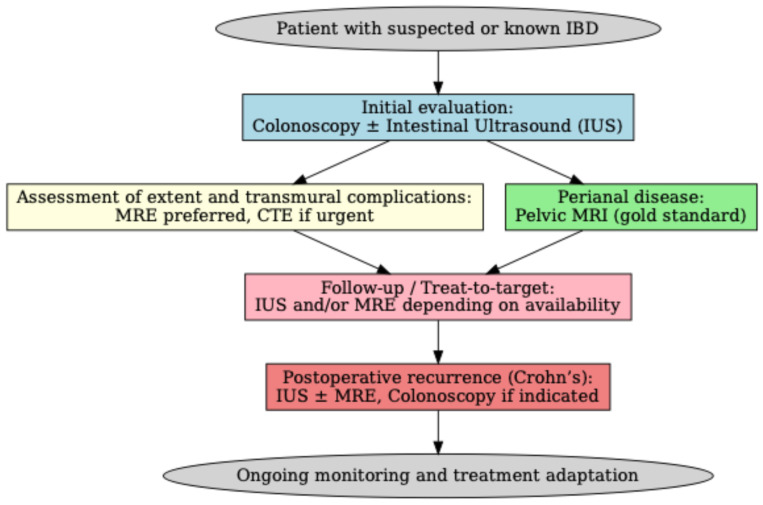
Algorithm for the use of imaging in IBD. Colonoscopy ± intestinal ultrasound is recommended at initial evaluation. Magnetic resonance enterography is preferred for disease extent and complications, with computed tomography enterography reserved for urgent settings. Pelvic MRI is the gold standard for perianal disease. IUS and/or MRE are suited for follow-up and treat-to-target monitoring, while postoperative recurrence in CD is best assessed with IUS, MRE, and colonoscopy if indicated.

**Figure 2 diagnostics-15-02457-f002:**
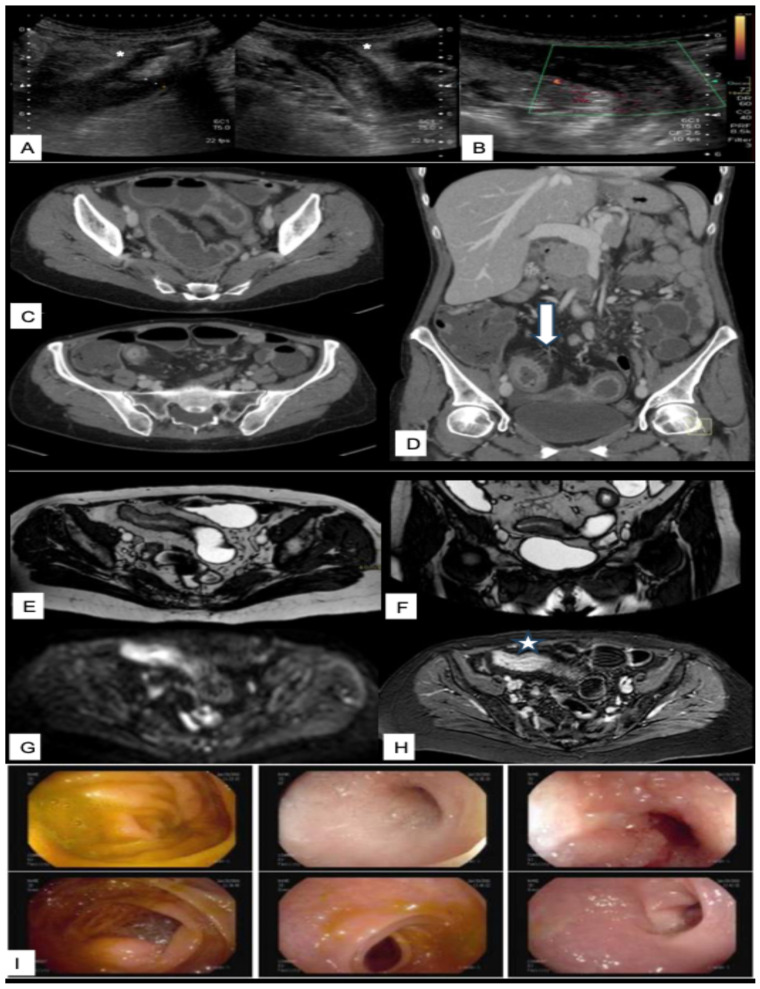
Multimodal imaging assessment of terminal ileal Crohn’s disease. Ultrasound (**A**) shows a concentric thickening of the terminal ileal loop (asterisk) with increased vascularity on color Doppler (**B**), suggesting active inflammation. Contrast-enhanced CT (**C**,**D**) reveals intense mucosal enhancement and submucosal edema in the terminal ileum, associated with mesenteric vascular engorgement (arrow). A segmental caliber reduction is observed at the terminal ileum, with upstream bowel dilatation extending to another less distensible ileal loop in the left iliac fossa. MRI axial (**E**) and coronal (**F**) T2-weighted sequences (confirms a 10 cm segment of concentric thickening of the terminal ileum; diffusion-weighted imaging (**G**) reveals limited restriction. Post-contrast T1-weighted sequence (**H**) demonstrate mild layered enhancement, with early mucosal-submucosal enhancement and delayed enhancement of outer layers, consstent with a fibrotic component (star). Ileocolonoscopy (**I**) confirms an ulcerated ileocecal valve (valicable with difficulty) and a non-traversable substenotic ulcerated segment, with hyperemic mucosa, erosions, and serpiginous ulcers involving two-thirds of the circumference.

**Figure 3 diagnostics-15-02457-f003:**
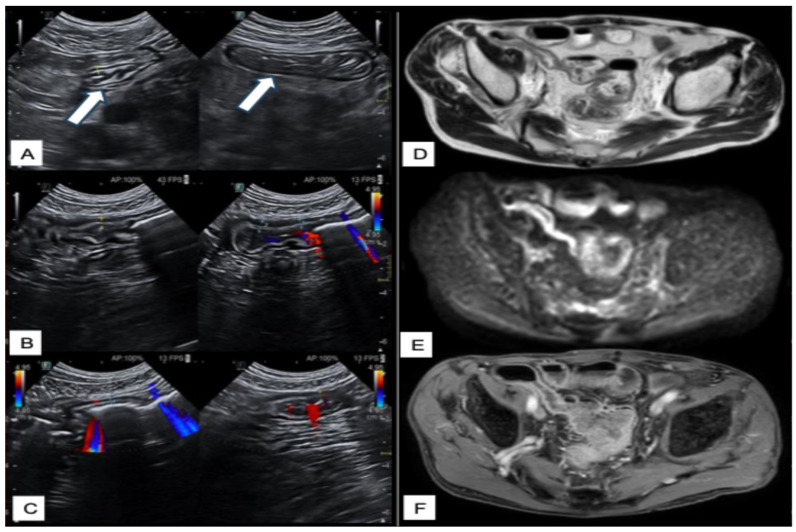
Multimodal imaging assessment of fibrostenotic Crohn’s disease of the terminal ileum. Ultrasound (**A**) of the right iliac fossa reveals a bowel loop referable to the terminal ileum, with segmental mural thickening, preserved stratification, regular margins (arrows), and no increased vascularity on color Doppler (**B**,**C**). T2-weighted MRI (**D**) confirms a concentric thickening in the terminal ileum. The involved bowel wall demonstrates a moderate degree of luminal stenosis. Diffusion-weighted imaging (**E**) reveals minimal restriction. Following gadolinium administration (**F**), enhancement is mild and primarily localized to the mucosal-submucosal layers, suggesting a predominant fibrotic component.

**Figure 4 diagnostics-15-02457-f004:**
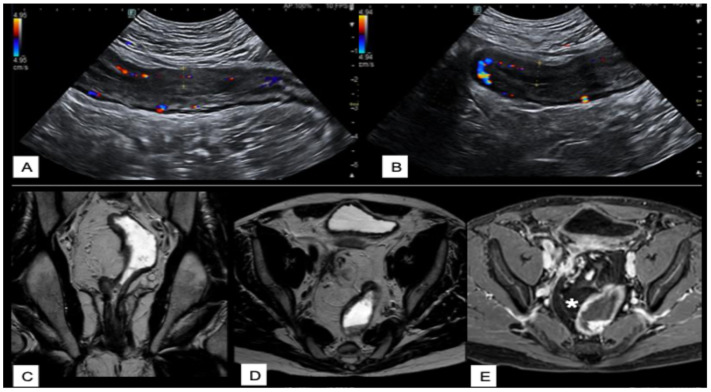
Imaging findings in a case of pouchitis with involvement of the afferent ileal loop. Ultrasound (**A**,**B**) of the right iliac fossa shows thickening of the afferent ileal loop proximal to the pouch, with a wall thickness of approximately 8 mm, hypoechoic wall stratification, and marked vascular signal on color Doppler, consistent with active inflammation. Contrast-enhanced MRI confirms bowel wall thickening on coronal (**C**) and axial (**D**) T2-weighted sequences, associated with post-contrast enhancement onT1-weighted images (**E**). There is also signal inhomogeneity of the peripouch fat (asterisk), indicating surrounding mesenteric inflammation. These findings are suggestive of active inflammatory pouchitis with involvement of the afferent limb and adjacent mesenteric fat, potentially indicating a more extensive or complicated disease.

**Figure 5 diagnostics-15-02457-f005:**
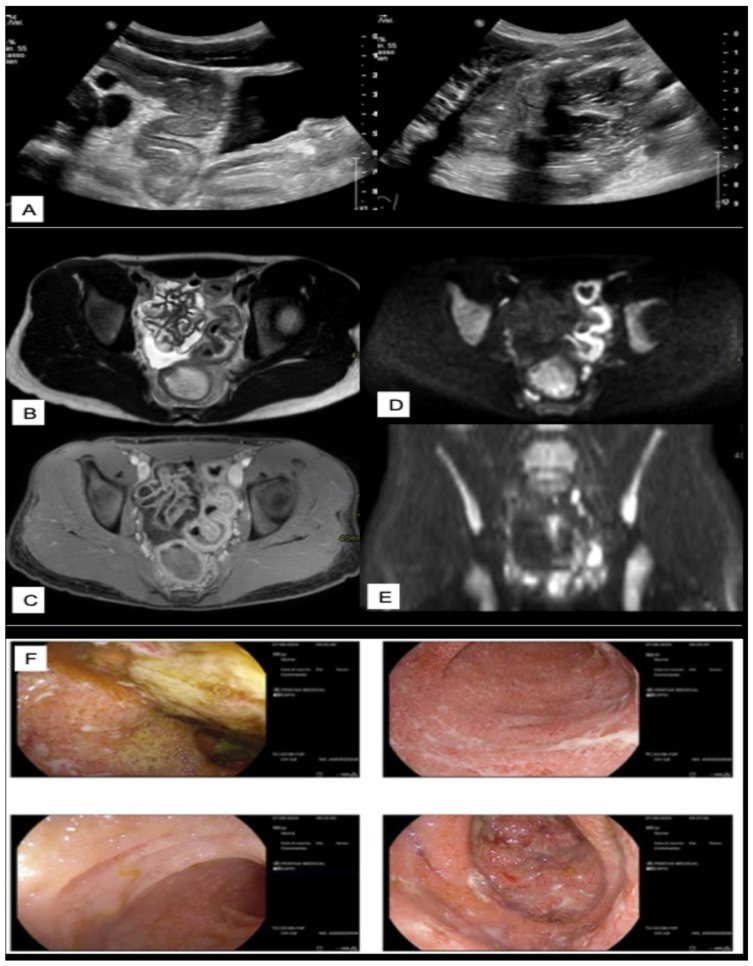
Multimodal imaging and endoscopic correlation in active ulcerative colitis with sigmoid predominance. Ultrasound (**A**) reveals diffuse, marked thickening of the colonic wall, particularly in the sigmoid colon, consistent with inflammatory changes. Surrounding mesenteric fat shows increased echogenicity and inhomogeneity, indicating active colitis. Axial T2-weighted sequence (**B**) on MRI confirms colonic wall thickening, especially in the sigmoid, with prominent post-contrast enhancement on T1-weighted images (**C**) and marked diffusion restriction (**D**,**E**) and. adjacent pericolic fat also appears inhomogeneous, reflecting active perivisceral inflammation. Colonoscopy (**F**) demonstrates continuous mucosal involvement from the cecum to the rectum, characterized by diffuse erythema, edema, loss of vascular pattern, and the presence of linear and geographic ulcers findings consistent with extensive active ulcerative colitis.

**Table 1 diagnostics-15-02457-t001:** Overview of clinical, endoscopic, and combined scoring systems used to assess disease activity in ulcerative colitis and Crohn’s disease, including adult and pediatric indices. Each score is characterized by specific parameters and thresholds to support diagnosis, classification of severity, treatment decisions, and monitoring of disease progression or response to therapy.

Score Name	Disease	Type	Description
**Truelove and Witts Severity Index** [11]	UC	Clinical	Classifies UC severity into mild, moderate, or severe based on stool frequency, blood in stool, temperature, heart rate, hemoglobin, and ESR (or CRP).
**Mayo Score (Full Mayo Score)** [12]	UC	Clinical + Endoscopic	Composite index evaluating stool frequency, rectal bleeding, endoscopic findings, and physician’s global assessment; total score ranges from 0 to 12.
**Simple Clinical Colitis Activity Index (SCCAI)** [13]	UC	Clinical	Assesses disease activity based on stool frequency, urgency, blood in stool, general well-being, and extracolonic features; scores range from 0 to 19.
**Ulcerative Colitis Disease Activity Index (UCDAI)** [14]	UC	Clinical + Endoscopic	Includes stool frequency, rectal bleeding, mucosal appearance, and physician’s assessment; total score ranges from 0 to 12.
**Ulcerative Colitis Endoscopic Index of Severity (UCEIS)** [18]	UC	Endoscopic	Evaluates vascular pattern, bleeding, and erosions/ulcers during endoscopy; scores range from 0 to 8.
**Crohn’s Disease Activity Index (CDAI)** [15]	CD	Clinical	Composite index assessing stool frequency, abdominal pain, general well-being, complications, use of antidiarrheal agents, hematocrit, and body weight; scores < 150 indicate remission.
**Harvey–Bradshaw** Index (HBI) [16]	CD	Clinical	Simplified version of CDAI evaluating general well-being, abdominal pain, number of liquid stools, abdominal mass, and complications; scores ≤ 4 suggest remission.
**Perianal Disease Activity Index (PDAI)** [17]	CD	Clinical	Assesses perianal disease severity based on discharge, pain, sexual activity restriction, type of perianal disease, and degree of induration; scores > 4 suggest active disease.
**Crohn’s Disease Endoscopic Index of Severity (CDEIS)** [19]	CD	Endoscopic	Evaluates the presence and extent of ulcerations, stenosis, and mucosal lesions across different bowel segments; scores > 5 indicate active disease.
**Simple Endoscopic Score for Crohn’s Disease (SES-CD)** [20]	CD	Endoscopic	Assesses ulcer size, ulcerated surface, affected surface, and presence of narrowings in five bowel segments; scores range from 0 to 56.
**Rutgeerts Score** [21]	CD	Endoscopic	Used postoperatively to assess recurrence in the neoterminal ileum; scores range from i0 (no lesions) to i4 (diffuse inflammation with large ulcers and/or nodules/cobble and/or narrowing/stenosis).
**Pediatric Ulcerative Colitis Activity Index (PUCAI)** [22]	UC (Pediatric)	Clinical	Non-invasive index assessing abdominal pain, rectal bleeding, stool consistency, number of stools, nocturnal stools, and activity level; scores < 10 indicate remission.
**Pediatric Crohn’s Disease Activity Index (PCDAI)** [23]	CD (Pediatric)	Clinical	Evaluates abdominal pain, stool frequency, general well-being, weight, height, and laboratory markers; scores < 10 suggest remission.

**Table 2 diagnostics-15-02457-t002:** Summary of intestinal ultrasound (IUS) scoring systems for Crohn’s disease and ulcerative colitis.

Scoring System	Disease	Key Parameters	Imaging Findings	Clinical Utility
**SUS-CD (Simple Ultrasound Score for Crohn’s Disease)** [65]	CD	Bowel Wall Thickness (BWT), Color Doppler Signal (CDS)	Increased BWT, hyperemia	Assesses disease activity; correlates with endoscopic indices
**BUSS (Bowel Ultrasound Score)**	CD	BWT, CDS	BWT > 3 mm, increased vascularity	Evaluates treatment response; correlates with endoscopic findings
**IBUS-SAS (International Bowel Ultrasound Segmental Activity Score)** [47]	CD	BWT, Bowel Wall Stratification (BWS), CDS, Inflammatory Fat	Loss of stratification, increased vascularity, hyperechoic mesenteric fat	Standardized assessment across centers; high interobserver agreement
**UCS (Ultrasound Consolidated Score)**	CD	BWT, Symmetry, Peribowel Fat Echogenicity, CDS, BWS, Bowel Wall Echogenicity	Asymmetrical thickening, hyperechoic fat, increased vascularity	Comprehensive assessment; correlates with endoscopic scores
**MUC (Milan Ultrasound Criteria)** [25]	UC	BWT, Colonic Wall Flow	BWT > 3 mm, increased vascularity	Differentiates active from inactive disease; validated against endoscopy; evaluates treatment response; predicts risk of colectomy
**UC-IUS (Ulcerative Colitis Intestinal Ultrasound Score)** [68]	UC	BWT, CDS, Haustration Patterns, Fat Wrapping	Loss of haustration, increased vascularity, hyperechoic fat	Correlates with Mayo Endoscopic Sub-score; monitors disease activity
**SPAUSS (Simple Pediatric Activity Ultrasound Score)**	Pediatric UC	BWT, CDS across colonic segments	Segmental BWT increase, hyperemia	Assesses disease activity in pediatric patients; correlates with clinical indices
**Civitelli Index** [68]	UC	BWT, CDS	Increased BWT, enhanced vascularity	Quantitative measure of disease activity; applicable in adults and children

**Table 3 diagnostics-15-02457-t003:** CT Enterography Protocol.

Parameter	Details
**Fasting**	Patients should fast for at least 4–6 h prior to the examination to minimize residual gastric contents and reduce motion artifacts.
**Oral Contrast Agent**	Ingest 1.5–2 L of a neutral or low-density oral contrast agent (e.g., water, polyethylene glycol solution, low-concentration barium) over 45–60 min before scanning to achieve adequate small bowel distension.
**Patient Positioning**	The patient is positioned supine on the CT table. Scanning is typically performed in the supine position, but prone positioning may be used to redistribute bowel loops and improve visualization.
**Intravenous Contrast**	Administer 100–120 mL of iodinated contrast material intravenously at a rate of 3–5 mL/s. Scanning is performed during the enteric phase, approximately 60–70 s after injection, to optimize visualization of the bowel wall and mesenteric vasculature.
**Scan Range**	Acquire images from the diaphragm to the symphysis pubis to encompass the entire small bowel and adjacent structures.
**Slice Thickness**	Utilize thin-slice acquisition (0.625–1.25 mm) with multiplanar reconstructions (axial, coronal, and sagittal) for detailed assessment of the bowel wall and surrounding tissues.
**Use of Antispasmodics**	Consider administration of antispasmodic agents (e.g., glucagon or hyoscine butylbromide) to reduce bowel peristalsis and motion artifacts, enhancing image quality.
**Post-Processing**	Perform multiplanar reconstructions and, if necessary, 3D volume rendering to evaluate the extent of disease, identify complications such as fistulas or abscesses, and assist in surgical planning.

**Table 4 diagnostics-15-02457-t004:** Main MRI sequences in IBD and their clinical relevance. Abbreviations: DWI = Diffusion-Weighted Imaging; ADC = Apparent Diffusion Coefficient; SSFP = Steady-State Free Precession.

MRI Sequence	Technical Principle	Imaging Appearance	Clinical Relevance in IBD
**T1-weighted (pre/post-contrast)**	Short TR/TE; fat appears bright, fluid dark. After gadolinium, enhances vascularized tissues.	Inflamed bowel wall shows post-contrast enhancement, often layered (mucosa/submucosa).	Detects mural hyperenhancement, ulcers, and stratification; useful for assessing active inflammation and differentiating fibrotic vs. inflammatory changes.
**T2-weighted (with fat suppression)**	Long TR/TE; fluid appears bright, fat suppressed for better contrast.	Bowel wall edema and ulcers appear as hyperintense areas; lumen fluid is also bright.	Highlights mural edema and inflammatory changes; sensitive for acute disease activity.
**DWI**	Sensitive to restriction of water molecule motion; high signal on high b-value images.	Inflamed segments show restricted diffusion (bright signal, low ADC values).	Identifies active inflammation even without contrast; useful for treatment monitoring.
**Balanced SSFP**	Gradient echo sequence with steady-state free precession; high signal from fluids.	Provides bright depiction of intraluminal fluid and bowel wall in real time.	Useful for assessing bowel motility, luminal narrowing, strictures, and overall loop anatomy.

**Table 5 diagnostics-15-02457-t005:** Recommended technical parameters for magnetic resonance enterography (MRE) in inflammatory bowel disease, including patient preparation, imaging acquisition, and sequence-specific settings. Parameters are adapted from ECCO–ESGAR guidelines and recent consensus recommendations to ensure optimal image quality and reproducibility. Abbreviations; HASTE = Half-Fourier Acquisition Single-shot Turbo Spin Echo; SSFSE = Single-Shot Fast Spin Echo; SSFP = Steady-State Free Precession; TrueFISP = True Fast Imaging with Steady-State Precession; FIESTA = Fast Imaging Employing Steady-State Acquisition; DWI = Diffusion-Weighted Imaging; EPI = Echo Planar Imaging; FOV = Field of View; TR = Repetition Time; TE = Echo Time; PEG = Polyethylene Glycol; IV = Intravenous.

Technical Parameter	Recommended Specification
**Patient Preparation**	
Fasting	4–6 h prior to the exam
Oral Contrast	Approximately 1000–1500 mL hyperosmolar oral contrast solution (e.g., Mannitol 2.5%, PEG), administered 45–60 min before imaging
Antiperistaltic Medication	Hyoscine Butylbromide, IV 20 mg, or Glucagon 1 mg IV (unless contraindicated)
Intravenous Contrast	Gadolinium-based contrast agent (0.1 mmol/kg)
Coil	Multichannel phased-array torso/body coil
Magnetic Field Strength	1.5 Tesla or 3 Tesla
Slice Thickness	3–5 mm
Slice Gap	0–1 mm
Imaging Planes	Coronal and Axial mandatory; Sagittal optional
Acquisition Technique	Breath-hold or free-breathing with respiratory triggering
Total Scan Duration	Approximately 25–35 min
**MRI Sequences and Parameters**	
**Cor T2-w (HASTE/SSFSE)**	
Purpose	Overview, bowel distension, fluid detection
TR/TE	>1000 ms/80–120 ms
Matrix	256–320 × 256–320
FOV	350–400 mm
Slice Thickness	3–5 mm
**Ax and Cor T2-w (Fat-Suppressed)**	
Purpose	Detection of bowel wall edema/inflammation
TR/TE	>1000 ms/80–120 ms
Matrix	256–320 × 256–320
FOV	300–400 mm
Slice Thickness	3–4 mm
**Balanced SSFP (TrueFISP/FIESTA)**	
Purpose	Identification of bowel motility, strictures, and fistulas
TR/TE	Shortest possible (<5 ms)
Flip angle	45–90°
Matrix	256 × 256
FOV	300–400 mm
Slice Thickness	4–6 mm
**Diffusion-Weighted Imaging (DWI)**	
Purpose	Identification of active inflammation
b-values	Typically 50, 600, and 800 s/mm^2^
Matrix	128–192 × 128–192
Slice Thickness	4–5 mm
Acquisition	Single-shot EPI
**Pre and post-contrast 3D T1-weighted Gradient Echo**	
Purpose	Evaluate enhancement pattern, mural stratification, ulcers
TR/TE	<5 ms/1–2 ms
Flip angle	10–15°
Matrix	256–320 × 192–256
FOV	350–400 mm
Slice Thickness	3 mm (with interpolation)
Phase	Pre-contrast, arterial (20–25 s), enteric (~45 s), and portal venous (~70 s) phases

**Table 6 diagnostics-15-02457-t006:** Imaging modalities tailored to key clinical scenarios in inflammatory bowel disease (IBD), with preferred techniques, main strengths, and limitations. The table emphasizes a scenario-driven approach to optimize diagnostic accuracy and clinical decision-making.

Clinical Scenario	Preferred Imaging Modality	Key Strengths	Limitations
**Assessment of strictures (fibrotic vs. inflammatory** **)**	MRE ± CEUS, CTE if urgent	MRE: superior for mural edema, stratification, fibrosis vs. inflammation; CEUS quantifies vascularity	CTE uses radiation; MRI limited availability
**Detection of fistulae (entero-enteric, entero-vesical, entero-cutaneous)**	MRE, pelvic MRI	High soft-tissue contrast; maps fistula tracts; defines complexity	Requires expertise; limited in acute emergencies
**Perianal fistulae and abscesses**	Pelvic MRI	Gold standard; precise classification (Parks); detection of abscesses	Cost, exam duration
**Detection of abscesses (intra-abdominal)**	MRE or CTE	Excellent sensitivity for fluid collections and inflammatory masses	MRI availability; CT involves radiation
**Postoperative recurrence (Crohn’s disease)**	IUS and MRE	Non-invasive monitoring; Rutgeerts score correlation; detects anastomotic recurrence	May miss subtle mucosal lesions
**Disease monitoring in UC**	IUS, CEUS, MRI	Non-invasive follow-up; correlates with endoscopic activity	Mild superficial disease may escape detection
**Pediatric patients**	IUS and MRE (no radiation)	Safe for repeated follow-up; good correlation with endoscopy	Requires cooperation and expertise

## Data Availability

No new data were created or analyzed in this study. Data sharing is not applicable to this article.

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
