# Peer review of "The Role of Imaging in Inflammatory Bowel Diseases: From Diagnosis to Individualized Therapy"

_diagnostics, 2025, doi:10.3390/diagnostics15192457_

Round 1
Reviewer 1 Report
Comments and Suggestions for Authors
This paper provides a comprehensive and exhaustive (if not exhausting!) review of all aspects of ultrasound in the diagnosis, assessment, prognosis, and monitoring of IBD. I am not enough of an expert in ultrasound technology to comment on the details of technique, but my overall impression is that this report has not clearly identified its audience. Is it aimed at clinical gastroenterologists to help them understand the uses of various imaging techniques for different clinical indications and scenarios? Or is it instead a detailed instruction manual for radiologists and technicians regarding patient positioning, contrast dosing, iv infusion rates, slice thickness, etc.? It seems to be trying to accomplish both purposes in one paper.
If on the one hand, it sought to explain the principles of various imaging techniques to clinicians, it could provide more of a basic glossary of terms like T1- and T2-weighted MRIs, color Doppler ultrasound, etc. Even more to the point for clinicians, it could be organized around various clinical scenarios and indications instead of around different imaging modalities. Alternatively, even procedure-oriented tables could have columns more explicitly stating what indications to procedures are best suited for.
For example, we could be better informed what modalities are best for stricture assessment (fixed versus inflammatory), detection of fistulae and abscesses according to location (ileal versus perianal), or detection of postoperative recurrence of Crohn’s disease, etc.
Similarly, clinicians might appreciate a focus on particular practice populations [e.g.,Hoerning A et al. Ultrasound in pediatric inflammatory bowel disease—a review of the state of the art and future perspectives. Children (Basel). 2024 Jan 25;11(2):156 doi: 10.3390/children11020156]. Or they might like to learn of the latest AGA guidelines on the uses of intestinal ultrasound [e.g., Chavennes M et al. AGA clinical practice update on the role of intestinal ultrasound in inflammatory bowel disease. Clin Practice Updates 2024;22(9):1790-5].
Clearly, it is the province of authors and not reviewers to decide what studies they want to do; but as a keenly interested reader, I would wish these authors had turned their Herculean energies to a report that would be more useful (and readable) for particular audiences.
Author Response
We thank the Reviewer for the thoughtful and constructive comments, which have greatly helped us to improve the clarity, focus, and clinical relevance of our manuscript. We appreciate this insightful suggestion and fully agree that clarifying the intended readership and better addressing clinical scenarios strengthens the manuscript. Accordingly, we have revised the text as follows:
- Target audience clarified: In the Introduction we now explicitly state that this review is aimed at both gastroenterologists and radiologists. We emphasize the dual purpose of bridging technical aspects with clinical applications, thereby fostering interdisciplinary collaboration in IBD care.
- Glossary of imaging terms: To facilitate understanding for non-radiologist readers, we have added a concise explaination of key imaging concepts, including T1- and T2-weighted MRI sequences, diffusion-weighted imaging, color Doppler ultrasound, and contrast-enhanced ultrasound.(Table 4)
- Clinical scenario–based organization: In addition to the section organized by imaging modality, we have added a new table (Table 6) that summarizes the preferred imaging techniques for the most relevant clinical scenarios in IBD (e.g., differentiation of fibrotic vs inflammatory strictures, detection of fistulae and abscesses, evaluation of perianal disease, and detection of postoperative recurrence).
- Updated references: As recommended, we have integrated recent authoritative references, including the AGA Clinical Practice Update on intestinal ultrasound (Chavannes et al., 2024) and the state-of-the-art review on pediatric IBD (Hoerning et al., 2024), while clarifying that the current work does not primarily aim to address pediatric populations. It should be noted, however, that the present review does not primarily aim to analyze pediatric populations, which have already been comprehensively addressed in dedicated works.
Reviewer 2 Report
Comments and Suggestions for Authors
Dear Editor
This is a well written review article regarding imaging studies in IBD. The followings are my comments.
1. The authors reviewed multiple imaging modalities in IBD. Could they propose an algorithm outlining the optimal timing and indications for each imaging modality in IBD?2. Regarding the use of radiomics and artificial intelligence in IBD assessment, could the authors elaborate on whether these techniques have a role in the initial diagnosis of IBD, particularly in differential diagnosis, as well as their potential role in predicting prognosis?
3. As patients with IBD carry an increased risk of colonic or small intestinal malignancy, could the authors expand their discussion to address this important point?
Author Response
We thank the Reviewer for the thoughtful and constructive comments, which have greatly helped us to improve the clarity, focus, and clinical relevance of our manuscript. Accordingly, we have revised the text as follows:
- We have added a practical, scenario-driven algorithm outlining the optimal timing and indications for each imaging modality in IBD (see Figure 1).
- In the revised version of the manuscript, we have expanded the section on radiomics and AI to clarify their potential role not only in disease activity assessment but also in the initial diagnosis and differential diagnosis of IBD (see section 3.6).
- We have revised the manuscript to include a dedicated paragraph in the Discussion (Section 7), where we address the increased risk of colorectal cancer in ulcerative colitis and the risk of small bowel adenocarcinoma in Crohn’s disease.
Reviewer 3 Report
Comments and Suggestions for Authors
This manuscript provides a comprehensive and timely review of advances in imaging techniques for inflammatory bowel disease (IBD), particularly touching on emerging developments such as AI and radiomics. The content is clinically meaningful and will be of interest to both researchers and practicing clinicians, as it highlights the potential of imaging to guide personalized treatment strategies. The review is well-structured and contains substantial value for the field. However, there are some concerns about the article.
- Inclusiveness of Imaging: Modalities. While the MRI-focused discussion is detailed and informative, non-invasive modalities such as intestinal ultrasound and capsule endoscopy are relatively underrepresented. Given the high practicality of intestinal ultrasound in routine practice and the unique strengths of capsule endoscopy in small bowel evaluation, more balanced coverage of these modalities would strengthen the manuscript.
- Cost-Effectiveness and Implementation Considerations: Diagnostic performance should be contextualized alongside cost, radiation exposure, examination time, and invasiveness, as these factors critically influence adoption in clinical practice. For example, while MRI and PET/MRI provide excellent diagnostic power, their high cost limits widespread implementation. Even a concise comparative discussion or summary table would significantly enhance the value of the review for clinical decision-making.
- AI and Radiomics: Opportunities and Risks; While the discussion of AI applications is forward-looking, challenges such as the "black box problem" and the risk of misdiagnosis through algorithm dependence also warrant attention. A brief acknowledgment of these limitations would help temper expectations and foster a realistic perspective.
- Role of Imaging in IBD-U: Unclassified IBD (IBD-U) is not uncommon in clinical practice, and imaging plays an essential role in follow-up and disease characterization. Current practice emphasizes a multimodal approach combining intestinal ultrasound, MRI, and endoscopy. Incorporating this perspective would substantially increase the clinical relevance of the review.
Author Response
We thank the Reviewer for the thoughtful and constructive comments, which have greatly helped us to improve the clarity, focus, and clinical relevance of our manuscript. Accordingly, we have revised the text as follows:
- In line with the suggestion, we have expanded the relevant sections of the manuscript to provide more balanced coverage of capsule endoscopy (see section 5).
- In line with the suggestion, we have added a dedicated section discussing cost-effectiveness and implementation considerations, with specific reference to diagnostic performance in relation to cost, radiation exposure, examination time, and invasiveness. These aspects are now summarized in Figure 1 and Table 6, which provide a concise comparative overview of the main imaging modalities.
- In accordance with the suggestion, we have revised the section on AI and radiomics to include a dedicated paragraph highlighting the main challenges and risks associated with their clinical implementation. Specifically, we now discuss the “black box” problem of deep learning models, the risk of misdiagnosis through algorithmic dependence, the lack of external validation across heterogeneous populations and imaging protocols, and the potential biases arising from non-representative training datasets. We emphasize the need for explainable AI approaches, external validation, and regulatory oversight before widespread clinical adoption. This addition provides a more balanced perspective, acknowledging both the opportunities and the limitations of AI in IBD management (see section 3.6).
- We have addressed this point by adding a dedicated paragraph (Section 6) discussing the role of imaging in IBD-U.
Round 2
Reviewer 1 Report
Comments and Suggestions for Authors
The authors have invested tremendous effort in revising their manuscript, and in so doing, they have greatly enhanced its readability and usefulness to both clinical and radiologic audiences.
Reviewer 3 Report
Comments and Suggestions for Authors
The authors have revised their manuscript, and no further comments are available.